# Immune-Mediated Neuropathies: Pathophysiology and Management

**DOI:** 10.3390/ijms24087288

**Published:** 2023-04-14

**Authors:** Abhishek Shastri, Ahmad Al Aiyan, Uday Kishore, Maria Elena Farrugia

**Affiliations:** 1Central and North West London NHS Foundation Trust, London NW1 3AX, UK; 2Department of Veterinary Medicine, UAE University, Al Ain P.O. Box 15551, United Arab Emirates; 3Department of Neurology, Institute of Neurological Sciences, Southern General Hospital, Glasgow G51 4TF, UK

**Keywords:** autoantibodies, demyelination, inflammation, immunity, neuropathy

## Abstract

Dysfunction of the immune system can result in damage of the peripheral nervous system. The immunological mechanisms, which include macrophage infiltration, inflammation and proliferation of Schwann cells, result in variable degrees of demyelination and axonal degeneration. Aetiology is diverse and, in some cases, may be precipitated by infection. Various animal models have contributed and helped to elucidate the pathophysiological mechanisms in acute and chronic inflammatory polyradiculoneuropathies (Guillain–Barre Syndrome and chronic inflammatory demyelinating polyradiculoneuropathy, respectively). The presence of specific anti-glycoconjugate antibodies indicates an underlying process of molecular mimicry and sometimes assists in the classification of these disorders, which often merely supports the clinical diagnosis. Now, the electrophysiological presence of conduction blocks is another important factor in characterizing another subgroup of treatable motor neuropathies (multifocal motor neuropathy with conduction block), which is distinct from Lewis–Sumner syndrome (multifocal acquired demyelinating sensory and motor neuropathy) in its response to treatment modalities as well as electrophysiological features. Furthermore, paraneoplastic neuropathies are also immune-mediated and are the result of an immune reaction to tumour cells that express onconeural antigens and mimic molecules expressed on the surface of neurons. The detection of specific paraneoplastic antibodies often assists the clinician in the investigation of an underlying, sometimes specific, malignancy. This review aims to discuss the immunological and pathophysiological mechanisms that are thought to be crucial in the aetiology of dysimmune neuropathies as well as their individual electrophysiological characteristics, their laboratory features and existing treatment options. Here, we aim to present a balance of discussion from these diverse angles that may be helpful in categorizing disease and establishing prognosis.

## 1. Introduction

Immune-mediated neuropathies are a heterogenous group of disorders affecting the peripheral nervous system (PNS), due to dysregulation of the immune system. The course of the disease is varying and can be acute, sub-acute, chronic or relapsing-remitting. Such peripheral neuropathies are generally characterised by progressive muscle weakness and accompanied by sensory deficits and can be caused by immune response against autoantigens in the PNS. There are several types of immune-mediated neuropathies such as Guillain–Barre syndrome (GBS) and its subtypes, chronic inflammatory demyelinating polyradiculoneuropathy (CIDP), multifocal motor neuropathy (MMN) and paraproteinemic neuropathies. Given that the main pathophysiology is immune-mediated, it would mean that with early detection, potential treatment of neuropathies is possible. This review aims to highlight the pathophysiological mechanisms, immunological processes and management of these conditions.

## 2. Guillain–Barre Syndrome

Guillain–Barre syndrome (GBS) is an inflammatory polyneuropathy characterised by a progressive flaccid ascending and symmetric muscle weakness associated with motor features and with or without sensory symptoms. Guillian, Barre and Strohl were the first to define this condition as a progressive motor deficit including more than one limb, often symmetric, associated with hyporeflexia or areflexia and with the maximum deficit being attained within four weeks of onset [1]. They also described the characteristic features of cytoalbuminologic dissociation of the CSF (refers to the presence of a normal cell count but with high levels of protein) [2]. The incidence ranges from 0.89 to 1.89 cases per 100,000 person-years [3,4], with a 40% higher rate in males and a 50% increase for every 10-year age increase [5]. The diagnosis is based on the clinical presentation and neurological findings supported by CSF and electrophysiological features [2]. Electrophysiological characteristic findings include prolonged distal motor latency (Distal latency is defined as the time from the stimulus of the nerve to the beginning of the response recorded when the distal-most (towards the muscle fibre) site of a nerve is stimulated. The response may be motor or sensory, depending on the nerve involved) [6]; prolonged or repeater F-waves (a F-wave is obtained via supramaximal stimulation of the motor nerve in the distal site. This causes the stimulus to reach the proximal (toward the neurons of the spinal cord) motor neurons and they fire, leading to a stimulus, which is measured as the F-wave) [7] along with conduction blocks [8]; and increased A-waves (an A-wave is recorded during routine F-wave studies. It appears during the acute stage of AIDP. The amplitude of the A-wave is shorter as compared to the F-wave) [9]).

The features of autonomic dysfunction such as arrhythmias, postural hypotension and bladder disturbances are also associated with GBS, and their effective management helps in determining the prognosis [10,11,12,13,14,15] Moreover, up to a third of patients usually require mechanical ventilation due to significant bulbar and respiratory muscle weakness. Approximately 20–50% of patients are left with residual and significant disability. The rate of mortality differs among centres of care but ranges between 5 and 10% [16,17,18,19,20,21]. Different subtypes of GBS are recognised and will be dealt with next.

Clinically, GBS can be classified into classic GBS (acute, flaccid paralysis of all four limbs), pharyngeal-cervical-brachial weakness (weakness of the neck, arms and oropharynx, or arm areflexia/hyporeflexia in the absence of limb weakness), paraparetic GBS (leg weakness and leg areflexia/hyporeflexia and the absence of arm weakness) and bilateral weakness with paraesthesias (facial weakness and limb areflexia/hyporeflexia, with the absence of limb weakness and hypersomnolence) [22]. Electrophysiological/nerve conduction studies help classify GBS into demyelinating (acute inflammatory demyelinating polyradiculoneuropathy: AIDP) and axonal (acute motor axonal neuropathy: AMAN and acute motor sensory axonal neuropathy: AMSAN) forms [23]. Nerve conduction studies (NCS) are helpful in understanding the type of injury, localising the injury and determining the severity of injury. It is performed for both motor and sensory nerve studies. In NCS, the nerve is stimulated electrically, and the sensory and motor responses are recorded. The motor response is called compound muscle action potential (CMAP), while the sensory response is called sensory nerve action potential (SNAP). The amplitude correlates with the number of motor nerve axons in CMAP, and with the number of sensory nerve axons in SNAP. In axonal loss, the amplitude of CMAP and SNAP are reduced, with mild slowing of the conduction velocity and the prolongation of distal latency (a measure for the velocity of distal conduction). In demyelination, the conduction velocity is slowed and/or with the prolongation of distal latency, while changes in amplitude may occur secondary to axonal loss. An increase in CMAP duration of more than 15% i.e., slowing of conduction velocity resulting in abnormal dispersion, is referred to as temporal dispersion. A conduction block and temporal dispersion signify demyelination [24,25].

Progression of GBS can be rapid, with maximum disability reached within 2 weeks, and one in five patients may develop respiratory failure [26,27]. The involvement of the autonomic nervous system (cardiac arrythmias and fluctuating blood pressure) contributes to mortality, which may occur in 3–10% of patients [28]. The clinical symptoms of GBS patients usually plateau after the initial progressive phase with 60–80% recovering in 6 months, while relapses are observed in 2–5% of patients [29,30].

The pathophysiology of GBS is the result of autoimmune processes, whereby both cell-mediated and humoral responses come into play. The immunobiology involves T- and B-cell responses, complement system (discussed separately), antiganglioside antibodies and molecular mimicry [31]. Molecular mimicry deals with antecedent infection with micro-organisms wherein identical epitopes are shared incidentally between microbial structures and nerve structures, leading to a cellular immune response. Microbes that have been significantly associated with GBS are *Campylobacter jejuni, Mycoplasma pneumonia*, *Haemophilus influenza*, Epstein–Barr virus, cytomegalovirus (CMV), varicella zoster virus and influenza virus [15,32,33,34], out of which the most frequently associated organism is *C. jejuni* [35].

### 2.1. Acute Inflammatory Demyelinating Polyradiculoneuropathy

The most common subtype of GBS is acute inflammatory demyelinating polyradiculopathy (AIDP) [36]. Early reports of nerve biopsy or autopsy findings supported an immune-mediated aetiology [37]. The pathological changes observed in AIDP involve segmental demyelination and infiltration of T-lymphocytes (may be minimal at times) and macrophages [37,38] Experimental allergic neuritis (EAN), the experimental animal model of AIDP, revealed that both cellular and humoral immune responses contributed to the pathogenesis [39]. EAN is induced by active immunization of the Lewis rat with the bovine peripheral nerve myelin, and pathogenesis is due to an immune response against any of the myelin glycoproteins P0, P2 or PMP22 [40,41,42,43]. The role of T-cells is further established by the development of EAN brought about by passive or adoptive transfer of myelin-protein specific T-cell lines [44,45,46,47].

The presence of nerve oedema, perivenular lymphocytic infiltrates and macrophage-mediated demyelination is reminiscent of the pathological findings in GBS patients. The model shows CD4^+^ T-cell-mediated response, with T helper type I (T_h_1) responding against the myelin proteins along with an increased number of T_h_1 cells in rat lymphoid organs, secreting interferon (IFN)-γ (T_h_1 cells are sub-sets of CD4^+^ T-cells that produce cytokines such as IFN-γ, TNF-α and IL-10. These are involved in cell-mediated response by stimulating macrophages) [48]. IFN-γ localises in nerve roots and induces nerve damage [49] via macrophage activation [50,51,52] and by increasing the production of other pro-inflammatory cytokines such as tumour necrosis factor (TNF)-α and interleukin (IL)-1 [53]. The role of macrophages in GBS is poorly understood. The infiltration of macrophages due to the breakdown of the blood–nerve barrier and macrophage-associated demyelination has been observed in GBS patients [54]. Furthermore, macrophage-mediated secretion of proinflammatory chemokines and cytokines such as TNFα have also been observed in GBS patients, which also increases macrophage infiltration and myelin phagocytosis [54]. The CSF of GBS patients show the increased median concentration of proinflammatory cytokines IL8 and IL1, and chemokines as compared to healthy controls [55]. Recently, Zhu et al. [56] confirmed that T-cell immune response to myelin plays an important role in determining the susceptibility to EAN in Lewis rat strains. However, there is also a strong B cell response (antibody-mediated response) to a large number of myelin antigens and they play an additive role in T-cell-mediated responses [56] In a study by Brunn et al. [57], the authors concluded that B cells are considered to play a dual role of immunoregulation during the early phase and a pro-inflammatory role during the recovery phase of EAN.

The up-regulation of IFN-γ and TNF-α correlates with the maximum clinical severity while that of anti-inflammatory cytokines IL-4, IL-10 and transforming growth factor (TGF)-β correlate with recovery [48], suggesting that anti-inflammatory cytokines help improve the pathology. The amount of IFN-γ secreted by lymphocytes in EAN-induced Lewis rat strains is directly proportional to the clinical severity while the amount of TGF-β secreted is inversely proportional to the clinical severity [58]. However, mice deficient in IFN-γ also show increased severity of EAN. In this study, an increased activation of T_h_17 cells was found (T_h_17 cells are sub-sets of CD4^+^ cells that produce IL-17) [59]. Furthermore, levels of T_h_17 cells have been found to be higher in GBS patients [60]. Studies relating to TNF-α in EAN have led to interesting results. The inhibition or neutralisation of TNF-α by soluble TNF-α receptor shows the improvement in EAN [61,62]; TNF-α receptor-deficient mice show reduced clinical signs [63]. Lu et al. (2007) [64] reported more severe clinical signs in TNF-α receptor-deficient mice. Another murine study on TNF-α-deficient mice show reduced EAN, caused probably due to alteration in the balance of M1/M2 macrophages (M1 macrophages are ‘classically’ activated macrophages that have pro-inflammatory phenotype. M2 macrophages are ‘alternatively’ activated macrophages that usually have immunoregulatory roles) [59]. TNF-α and IL-1β have also been localised using immunohistochemistry on Schwann cell membranes [65]. The role of cytokines in GBS and EAN has been reviewed extensively by Lu and Zhu (2011) [66]. IVIG has also been shown to reduce circulating levels of proinflammatory cytokines TNFα and IL1β, and clinical improvement was correlated with reduced TNFα levels [67]. Interestingly, the usage of TNFα antagonists has been reported in several case studies to be associated with GBS pathogenesis; the exact mechanism is not fully known, but it is thought that TNFα antagonists may be causing increased susceptibility to latent infections, thereby making the autoimmune process worse [68,69]. Another interesting development is the up-regulation of erythropoietin in the PNS of rat model of EAN (Erythropoietin is a cytokine for erythrocyte precursors that leads to their differentiation and proliferation into erythrocytes. It is produced mainly by the kidneys) [70]. Erythropoietin has been found to play a beneficial role by leading to the production of TGF-β [71].

During the clinical course of EAN, there is a sequential up-regulation of chemokines (Chemokines are cytokines that help in the migration and activation of inflammatory cells such as phagocytes and lymphocytes) [72]. Chemokines such as macrophage inflammatory proteins (MIP)-1α and MIP-1β peak post-immunization, just before the maximum disease severity is attained, whereas the levels of monocyte chemoattractant protein (MCP)-1 and IFN-γ-inducible-protein-10 peak later [73].

Matrix metalloproteinases (MMPs) are found to be produced in EAN and GBS [74,75], especially MMP-9 and MMP-2, which are co-related with increasing severity and levels of pro-inflammatory cytokines [76,77,78]. MMPs are postulated to cause the destruction of nerve cells by lysis of the basement membrane, leading to migration of inflammatory cells towards nerve tissue. Interestingly, MMP-2 has also been found to have a physiological role in causing myelination in the PNS [79].

Toll-like receptors (TLRs), involved in innate immunity and the promotion of adaptive immunity, have also been shown to be involved in GBS and EAN (TLRs are pattern-recognition receptors of the innate immune system. These TLRs recognise molecular patterns called pathogen-associated molecular pattern (foreign/non-self) and damage-associated molecular pattern (altered self) to launch an immune response in the body) [80]. TLR2 and TLR6 expression is increased in EAN and GBS patients [81] while another study found an increase in TLR2, TLR4 and TLR9 expression in GBS patients [82]. TLR2 also promotes the differentiation and proliferation of T_h_17 cells [83], and the activation of T_h_17 cells is associated with EAN [84].

### 2.2. Axonal Types of Guillain–Barre Syndrome

The axonal types of GBS consist of acute motor axonal neuropathy (AMAN) and acute motor sensory axonal neuropathy (AMSAN). Antibodies to ganglioside-like moieties in peripheral nerves are found in the axonal type of GBS [31,85,86,87,88]. Gangliosides are a major constituent of a nerve cell and are expressed in the nervous system [89]. These sialic acid-containing glycosphingolipids help in the growth, development and repair of the nervous system [90] as well as the stability of interactions between neurons and glial cells [91]. Some of the gangliosides are GM1, GM2, GD1a, GD1b and GQ1b [92]. Anti-GM1, anti-GM1b, anti-GD1a and anti-GalNAc-GD1a antibodies have been found to occur when GBS is preceded by *Campylobacter jejuni* infection [93,94]. CMV is associated with anti-GM2 antibodies while *M. pnuemoniae* is associated with antibodies against galactocerebroside [94].

The requisites for molecular mimicry according to Ang et al. (2004) [89] include the epidemiological association between microbes and disease, the presence of cellular and humoral immunity, the identification of antigen-mimics, and the reproduction of disease in animal models, all of which are observed in GBS. The first case of such a concept of molecular mimicry was reported when it was found that the terminal structure of lipooligosaccharide (LOS) expressed on *C. jejuni* is identical to the terminal tetrasaccharide of GM1 ganglioside [95,96,97]. This was found to lead to the production of anti-GM1 antibodies that bind to GM1 on the nodes, leading to conduction blocks and degeneration [98]. This carbohydrate mimicry was later confirmed by Yuki et al. (1994) [99] where *C. jejuni* LOS sensitisation produced anti-GM1 antibodies as well as GBS pathology in rabbits.

Molecular mimicry in this manner has been found to be a major cause of GBS in developing countries [100]. The sialic acid composition on LOS has been shown to determine the type of T_h_ cell response [101]. In vitro studies show that neurite outgrowth is inhibited by antiganglioside antibodies [102]. The passive transfer of antibodies to gangliosides has been found to be pathogenic, and in the presence of complement has led to the development of animal models of disease [103,104,105,106].

The nodes of Ranvier contain high density of sodium channels [107]. Voltage-clamp studies show that antibodies of GBS patients alter potassium and sodium currents [108], with evidence that these antibodies not only co-localise with channels [109] but also directly impair the functioning of sodium channels in a complement-dependent manner [110], as well as the Cav 2.1 voltage-dependent calcium channel current [111,112]. Autoimmunity against neurofascin, which is involved in cell adhesion of axons and Schwann cells, have also been reported [113]. Other nodal proteins such as gliomedin and contactin have also been found to be antigens of target for GBS [114]. Devaux (2012) [115] found that antibodies to gliomedin cause neuropathy in Lewis rats and dismantle the organization and sodium cluster channels at the nodes. Furthermore, the clearance of anti-ganglioside antibodies at pre-synaptic motor nerve terminals via vesicular endocytosis decreased the complement activation in vitro and ex vivo, which offers protection against injury. The same, however, does not happen at the node of Ranvier, making it more vulnerable to complement attack [116].

#### 2.2.1. Acute Motor Axonal Neuropathy (AMAN)

A Chinese-American group jointly recognised that during summer epidemics of GBS, the primary problem was dysfunction of the motor axon in the absence of sensory nerve involvement or demyelinating features. This was termed as AMAN [117,118] (Figure 1). The pathophysiology in AMAN is different as it involves an antibody-mediated process against ganglioside antigens on the axolemma and very little lymphocytic infiltration [119]. The myelin sheath remains intact and macrophages invade the nodes of Ranvier at sites between the axon and Schwann cell axolemma [117]. The pathological process usually results in conduction blocks especially during the early stages of illness [23] and the axon rarely degenerates as the damage is localized distally [120]. This results in a rapid clinical decline of the patient but with subsequent rapid reversal of conduction failure, leads to improvement of neurological dysfunction [23,121]. According to Uncini and Kuwabara (2012) [122], this reversible conduction failure can be detected only by serial recordings, and thus, this must be included in the electrodiagnostic feature of AMAN. An imaging study performed both in and ex vivo showed the rapid recovery of distal motor nerve terminals when subjected to anti-ganglioside antibody and complement-mediated injury [123]. In vitro studies by Cencioni et al. (2009) [124] report the involvement of cellular immunity (T cell response to gangliosides) in the pathogenesis of AMAN. In AMAN, the ventral root is attacked, resulting in degeneration of the entire axon. The immunisation of animals with ganglioside GM1 induces an acute neuropathy that is histologically similar to AMAN [99]. Interestingly, GM1-specific antibodies are found to be associated with complement and leucocyte activation in AMAN [125].

#### 2.2.2. Acute Motor Sensory Axonal Neuropathy (AMSAN)

AMSAN is more common in Asia and South America. Antibodies involved are anti-GM1 and anti-GD1a antibodies [126,127]. This subgroup involves both the motor and sensory axons. The clinical course has rapid progression. Patients generally have a poor prognosis with a high mortality rate [128] and recovery may be slow. Glove-and-stocking type of sensory loss is observed [129]. Facial muscle weakness along with areflexia may also be observed [130]. The pathology is similar to AMAN but with the additional involvement of the sensory component. The myelin sheath remains intact, although unlike AMAN, there is destruction of the axon. For the diagnosis of AMSAN, there should be no evidence of demyelination [122]. Furthermore, in AMSAN, both the dorsal and ventral roots are involved. The infiltration of lymphocytes is minimal or absent [130]. Nerve conduction studies may even show absent responses. Reversible conduction failure has also been reported in AMSAN. The observations in AMAN with the involvement of sensory fibres, albeit to a lesser extent, have led to the argument that both AMAN and AMSAN are a continuum in axonal form of GBS with AMSAN being the more severe end of this spectrum due to additional sensory fibre involvement [131,132].

### 2.3. Miller–Fisher Syndrome (MFS)

The Miller–Fisher variant bears many similarities to GBS and is characterised by ataxia (disproportionate to the motor and sensory deficit), areflexia and ophthalmoplegia [133]. In the Miller–Fisher variant, up to 90% of patients have antibodies to GQ1b, which bind to cranial motor nerves [134,135]. The GQ1b ganglioside is present in the oculomotor, abducent and trochlear cranial nerves [136]. Molecular mimicry is observed in *Campylobacter jejuni* infection leading to MFS [99]. Electrophysiological changes include abnormality in sensory conduction [33], reduced sensory nerve action potential [87] and absent H-reflex (H-reflex study is usually conducted on the gastrocnemius and soleus muscles (back of the knee). Low electrical stimulation of sensory fibres of the muscle leads to H-reflex but with an increase in the strength of stimulus, H-reflex disappears and F-wave appears) [136]. The pathology of MFS, whether it is demyelinating or axonal, is not entirely known. Reversible conduction failure, observed more often in AMAN, has also been reported in a study, along with the absence of demyelination, leading to a proposal that MFS is also part of the spectrum of axonal GBS [137,138]. MFS is not solely a peripheral nerve disorder, with PET studies showing hypermetabolism in the central nervous system, mainly in the brainstem and cerebellum [139]. Transcranial magnetic stimulation studies show evidence of cerebellar dysfunction in MFS, which normalises when symptoms improve and resolve [140]. The prognosis is much better in MFS when compared to the other variants. Complete resolution may occur within six months [141], and this may be the reason immuno-modulating treatment modalities are ineffective [142].

### 2.4. Roles of Ganglioside Complexes, Complement System and SARS-CoV-2 in Guillain–Barre Syndrome

Gangliosides are expressed on both neuronal and glial cell membranes. The GD1a antibody is found to preferentially stain motor fibres and is thus associated more with AMAN, while GD1b antibodies have been found to preferentially stain sensory nerve fibres and are more associated with the sensory type of GBS [143]. GM1 moieties have been shown to be present on the nodes of Ranvier, Schwann cells and on axolemmal surface of mature myelinated nerve fibres [109]. Moreover, studies have found that anti-GM1 antibodies may preferentially bind glial or axonal GM1, thereby developing differential antibody titres and clinical symptoms [144].

Antibodies in GBS and MFS often bind to ganglioside complexes, formed by two or more gangliosides (Table 1). These complexes are present within the microdomains of lipid rafts [145]. The conformational epitopes expressed by these complexes differ from those expressed by individual gangliosides [146]. Antibodies to these complexes are more likely to activate the complement classical pathway and, therefore, are considered to be more pro-inflammatory. Anti-ganglioside complexes GD1a/GD1b and GD1b/GT1b are associated with severe disability in GBS [147] and complexes that include anti-GQ1b are expectedly more prevalent in MFS [148]. Antibodies to certain ganglioside complexes such as GD1a/GD1b and GD1b/GT1b complexes can help determine prognosis, and may be associated with more severe GBS, requiring ventilation [149]. Other ganglioside complexes in GBS include GM1 and GalNAc-GD1a complex [150]. *C. jejuni* LOS has been shown to cross-react with GM1/GD1a and GQ1b/GD1a complexes and may also induce the ganglioside complex formation (Table 1; [151]).

The complement system is part of the innate immune system and consists of more than 30 proteins present in serum. It consists of three different initiating pathways culminating into the formation of the membrane attack complex (MAC) that causes lysis of pathogenic or abnormal cells. The initiating pathways are the classical pathway (usually activated by the antigen-antibody complex recognition by C1q); alternative pathway (activated spontaneously by low level hydrolysis of C3 to C3(H_2_O) and the lectin pathway (activated by the recognition of microbial carbohydrate mannan by mannan-binding lectin). All the three pathways ultimately lead to the formation of C3 convertase, which cleaves C3 into C3a and C3b. This C3b is involved in the formation of C5 convertase, which cleaves C5 into C5a and C5b. C3a and C5a are anaphylatoxins. C5b associates with C6, C7, C8 and C9 to form MAC. The complement system is regulated by a number of proteins such as the C1 inhibitor, factor H, properdin, C4b-binding protein, CR1, CD55 and CD59 to name a few.

Complement system plays an active and key role in neuroinflammation and neurodegeneration in the central nervous system. Complement proteins have also been shown to be expressed and synthesized in the peripheral nerves [152]. Schwann cells are thought to be the main source of complement proteins [153,154]. Complement components such as C1q, C1r, C1s, C4 and C3 and its regulators such as CD59, C1 inhibitor, C4b-binding protein, CR1 and CD55 have been shown to be produced in the peripheral nerves (axon, Schwann cell, myelin and endoneurium) [152,155,156].

In GBS, the immune response is associated with complement activation, which can be detected in serum [157] and CSF [158]. The deposition of activated complement products have been observed on the surface of Schwann cells, which may indicate an antibody-mediated injury [159]. Antibodies to gangliosides can bind to the node of Ranvier and activate complement proteins [160]. Recent data by Susuki et al. (2012) [161] indicate that complement-mediated damage to nodes result in the development of neuropathies in rats. In this study, the intraneural injection of anti-GD1a/GT1b antibodies caused complement deposition on nodes and its disruption, and it was also observed that the injection of anti-GD1b antibodies caused complement-mediated nodal disruption, mainly in sensory fibres. The activation of the classical pathway of complement by myelin, in the absence of antibodies, is also observed [162]. Myelin also activates the alternative pathway [163]. The activation of complement leads to the deposition of MAC on myelin membranes [164]. In EAN, MAC deposition has been shown to occur before the onset of any clinical signs [165]. The expression of CD59 in EAN has also been reported [166], and being a regulator, this may be protective [167]. Nafamostat mesilate, a complement inhibitor, has been found to inhibit the deposition of complement and prevent the disruption of sodium channels in a rabbit model of GBS [168].

The mechanism of action of the anti-GQ1b antibodies in causing damage to the nerves in MFS is complement-mediated [169]. Therefore, the inhibition of complements may help in the treatment of MFS, which has been shown successfully in animal models [170] with the complement inhibitor rEV576 [104]. A study by Zitman et al. [171] in a mouse model showed that anti-GM1/GD1a and anti-GM1/GQ1b complexes cause complement-mediated damage at the motor nerve terminals. The involvement of complement and complement regulators in murine models of GBS has been reviewed by Willison et al. [172]. Anti-C1q (humanized) antibody ANX005 has been shown to inhibit the complement pathway and reduce symptoms in a mouse model of GBS, without side effects, thus laying the platform for further testing in humans [173]. The latest randomised-controlled trials involving eculizumab, an inhibitor of terminal component of the complement pathway showed that eculizumab is reasonably well-tolerated and effective in GBS, although the trials had small sample sizes, and thus, the results did not achieve statistical significance [174,175].

Recent data also indicate that SARS-CoV-2 (COVID-19)-related GBS cases are emerging [176]. In a collection of eleven case studies reported by Dalakas [177], COVID-19 was found to trigger GBS-like acute paralytic symptoms; such symptoms may even be the first manifestation of COVID-19 symptomatology. Ganglioside antibodies against GD1b, instead of GQ1b, were also detected in patients. Patients with elevated creatine kinase (a marker for muscle damage) were found to be responsive to IVIG, thus showing the benefits of exploring GBS-like symptomatology in patients with COVID-19. Furthermore, the early diagnosis of GBS in patients with COVID-19 is considered to be important, as the clinical course and progression correlate with the severity of disease and requirement of intensive care and mechanical ventilation [178]. A recent large-scale study also found that a replication-incompetent human adenovirus vector-based COVID-19 vaccine is associated with an increased risk of GBS [179]. The exact mechanism of action is poorly understood, but it is thought that the cell-surface antigen called as angiotensin-converting enzyme 2 (ACE2) and gangliosides are responsible for the interaction and uptake of SARS-CoV-2 [180,181]. ACE2 is also found on neurons and glial cells, and due to cross-reactivity and molecular mimicry, an immune-mediated destruction of myelin and axons are brought about [182] (Figure 2).

### 2.5. Management

The diagnosis of GBS is based on extensive clinical history and neurological examination, and supported by the results of electrophysiological studies and CSF findings. It is also important to rule out other differential diagnosis such as sarcoidosis, neuromyelitis optica, malignancy, stroke, vitamin deficiencies, nerve root infections, neuromuscular junction disorders etc. [27]. Routine laboratory investigations and lumbar puncture may also be useful in this regard. IVIG and plasma exchange are both equally effective treatments for GBS [183]. Any complication should be monitored, such as respiratory dysfunction, any progression of weakness in limbs, pain and autonomic dysfunction. Some of these features (impairment in respiratory function, requiring artificial ventilation, swallowing problems and autonomic dysfunction) may indicate intensive care unit admission. As part of the course of treatment, sometimes treatment-related fluctuations (worsening or progression of disease after an initial improvement) are observed. A retreatment with IVIG or plasma exchange may be performed, and if further deterioration is observed in 8 weeks or if three of more relapses are observed, then treatment is performed as CIDP. Physiotherapy, rehabilitation and psychological support are also important for the treatment and recovery of patients [27,184].

IVIG is thought to inhibit the Fc-mediated activation of immune cells, inhibit antiganglioside binding and inhibit complement activation, while plasma exchange is considered to remove toxic antibodies, complement system proteins and cytokines [183,185]. Understanding these mechanisms of actions further highlights the importance of these immune-mediators in GBS pathogenesis. The treatment of GBS may constitute plasmapheresis that usually includes at least five exchanges over a period of one week [33,186]. A recent Cochrane study found plasma exchange (PE) to be better than supportive care alone, along with an increased likelihood of full recovery after one year [187]. Alternatively, the patient may be treated with intravenous immunoglobulin (IVIG) usually with a regimen of 0.4 g/kg/day for five days [33]. In a Cochrane systematic review, a meta-analysis of five randomised trials that compared the use of IVIG started within two weeks of disease onset and PE showed no significant difference [183,188]. There was no association between the improvement in patients and treatment involving initial IVIG followed by PE [189]. Furthermore, IVIG showed better results in children than with supportive care alone [190]. IVIG is considered to act by providing anti-idiotype antibodies, by blocking active sites of Fc receptors on macrophages, inhibition of the complement cascade, modulation of T cell regulatory processes and saturation of neonatal Fc receptor [191,192,193,194]. Several promising trials involving FcRn inhibitors are ongoing and show promise by reducing autoantibodies especially in myasthenia gravis [192,195,196]. In contrast, corticosteroids have not been proven to be beneficial in the treatment of GBS [197,198]. This is difficult to explain in the context of a known immune-mediated process but corticosteroids have been postulated to interfere with the remyelination process and possibly aggravate damage of denervated muscle fibres [199,200] or inhibit macrophage repair processes [190]. Neurorehabilitation constitutes an integral part of the management, although a multidisciplinary approach is required for effective recovery of the patient with GBS [201]. Treatment options with complement inhibitors may become pivotal or indeed the mainstay of treatment in the future, especially if instituted early on in the process, thereby switching off complement-mediated destructive processes in the disease.

## 3. CIDP

In 1982, Dyck et al. [202] coined the term to summarise the clinical and pathological features of this condition. Clinical features for typical CIDP include developing progressive or recurrent symmetrical proximal and distal weakness associated with sensory dysfunction in the extremities over at least 2 months [203]. CIDP is presumed to be associated with the presence of antibodies that target the myelin sheaths of peripheral nerves and occurs in association with other conditions such as diabetes mellitus [204], connective tissue disorders, infections (for example hepatitis and HIV) or dysproteinaemias, including chronic lymphocytic leukaemia and B cell lymphoma. Patients present with similar symptoms as GBS [205] (Anonymous, 1991) but the clinical course may be variable by being monophasic, relapsing-remitting or chronic progressive [206]. However, it is erroneous to refer to CIDP as a mere chronic form of GBS as the two conditions are distinct. For example, CIDP responds to corticosteroids and other immunosuppressive treatment, unlike GBS, as will be discussed further below. Saperstein et al. (2001) [207] divides patients into four categories: (1) Chronic symmetrical proximal and distal weakness; (2) Distal symmetrical weakness with sensory loss (distal acquired demyelinating [DADS] neuropathy); (3) Asymmetrical weakness (multifocal motor neuropathy with conduction block [MMN]); (4) Asymmetrical motor and sensory deficits (multifocal acquired demyelinating sensory and motor [MADSAM] neuropathy). For the purpose of this review, we have dealt with the latter two subcategories separately (as discussed below), although we do recognise that they are a continuum within the same spectrum.

The clinical diagnosis of CIDP is based on CSF findings (raised protein level and less leucocyte level) in conjunction with typical electrophysiological features of prolonged F wave latency (due to involvement of proximal nerve segments) and slowing of conduction velocity of nerves (due to segmental demyelination) (Table 1). The diagnostic criteria as suggested by Koski et al. (2009) [208] include progressive polyneuropathy for at least 8 weeks without the presence of any genetic abnormality or serum paraprotein; and either (i) electrophysiological characteristics of F-wave latency or abnormal conduction velocity, or (ii) symmetric onset and weakness in all four limbs and proximal weakness in at least one limb. The diagnostic criteria as recommended by the European Federation of Neurological Societies and the Peripheral Nerve Society (EFNS/PNS) [203] include typical and other forms of CIDP (as mentioned above) for at least 2 months with typical CSF findings (raised protein and reduced leucocyte levels); decreased conduction velocity in at least one nerve; clinical improvement with treatment; and nerve biopsy findings of demyelination and/or remyelination. The interpretation of electrophysiological studies can be a challenge [209]. In a study by Allen et al., 2018 [209] on patients initially referred to have CIDP but were misdiagnosed, it was found that in nearly half of the patients, the initial interpretation was misleading. Some of the key features in this regard include amplitude-dependent slowing, which may also occur in ALS; amplitude-independent slowing which is also seen in patients with diabetes mellitus, and clinicians often ignore accurate electrophysiological findings in favour of CSF findings. MRI may be considered in adult patients who only fulfil possible electrophysiological criteria and if ultrasound is unavailable or non-contributory. CIDP is more likely if MRI shows enlargement and/or increased signal intensity of nerve roots [210]. Ultrasound may also be considered in adult patients who fulfil criteria for possible CIDP, and a diagnosis of CIDP is more likely if nerve enlargement is observed in at least two sites in proximal median nerve segments and/or the brachial nerve plexus [210].

Demyelination in proximal segments may result in distal axonal degeneration with drop-out of large myelinated nerve fibres [211]. Peripheral nerve biopsies show nerve oedema and lymphocytic infiltration with laboratory evidence, as in GBS, of humoral and cell-mediated immune responses involving CD4^+^ and CD8^+^ T cells, B cells and macrophages against a variety of myelin-derived antigens [163,212]. Rabbits, immunized with a single large dose of myelin, also develop EAN of a chronic and progressive relapsing course, with autoreactive T cells and antibodies directed against nerve antigens [213]. Increased T cell reactivity to gangliosides is found in the serum of CIDP patients [214]. The functioning of T regulatory cells has been found to be decreased [215]. Immunohistochemistry of nerve samples show increased T cells with γδ-receptors [216], CD4^+^ and CD8^+^ T cells [217,218].

Complement activation, especially C5a and terminal complement complex, has been found to be increased in the serum and CSF of patients with CIDP [219]. However, IVIG treatment in CIDP patients did not result in any change of C3a, C5a or terminal complement complexes when compared to the placebo group, thus showing that IVIG treatment does not modulate complement deposition or pathway [220]. An increase in matrix metalloproteinases, adhesion molecules and chemokines [221,222,223], which help in the migration of T cells into peripheral nerves [206], has been reported. Increased serum levels of TNF-α and IL-2 is observed in CIDP patients [224,225]. Increased CSF levels of IL-12 are also observed in CIDP [226].

B cells are antibody-producing precursors that are regulators of T-cell activation (CD4^+^ and CD8^+^ T cells) [227,228]. The Fc-gamma receptor IIB (FcγRIIB) expressed on B cells is required to maintain immunological tolerance, autoimmunity and regulation of immunoglobulins [229,230,231]. Impairment in function of FcγRIIB has been observed in CIDP [232], and partial restoration of FcγRIIB functioning is observed with IVIG treatment [233].

Antibodies against various glycolipids have been shown in a small proportion of CIDP patients (Table 1; [234]), especially myelin proteins PMP2 [235] and MPZ [236,237] and P0 [238]. The serum or IgG from CIDP patients when injected into rats has been shown to produce demyelination, indicating that there could be a role for autoantibodies in CIDP [239]. Over the last decade, antibodies against proteins located at the node of Ranvier have been discovered in a small proportion of CIDP patients [240]. These autoantibodies target neurofascin, contactin 1 and contactin-associated protein 1, which are present at the node and paranode regions of myelinated peripheral nerves. The pathological features include the dissection or dismantling of myelin at the paranode, absence of segmental demyelination and mechanism of pathogenesis not being inflammatory or driven by macrophages [241]. Due to these features, a different terminology is considered to fit the presentation better: nodo-paranodopathies. Clinical features show acute or subacute onset, severe disability at the nadir, predominantly distal motor weakness and severe sensory ataxia [242]. Electrophysiological features include reversible conduction failure, prolonged F-wave and distal motor latencies [242]. Most patients show the presence of IgG4 type of antibody, and one of the key features of this type of antibody is that it does not activate complements, thereby making treatment with IVIG less effective. Patients show a good response to rituximab and/or corticosteroids [242].

The prognosis and response to treatment are variable [233]. The disappearance of conduction blocks and remyelination are often associated with the recovery of function, and the return of muscle strength, often occurring 3–4 weeks following the acute monophasic demyelinating insult. The outcome is more favourable when CIDP adopts a relapsing-remitting course [212]. The age of onset plays an important role in determining the outcome, with juvenile-onset CIDP showing more relapses and better recovery than in onset at old age where less relapses and less recovery are observed [243].

CIDP is a treatable condition and responds particularly to corticosteroid treatment [198,244]. In the first instance, the patient with a significant neurological deficit is usually treated with IVIG. A systematic Cochrane review confirms the effectiveness of treatment using this modality [245]. IVIG may contribute to improving axonal functions and recovery [246], FcγRIIB expression and [232] and reduction in activation of T cells [218]. IVIG and corticosteroids have been found to be comparable in the treatment of CIDP [233,247]. Various immunosuppressive and immunomodulating agents have been utilised in the treatment of CIDP including azathioprine, cyclophosphamide, methotrexate, cyclosporin A, Mycophenolate mofetil, rituximab, and interferon α and β. All these agents are potentially toxic, and unfortunately, systematic reviews conclude that there is no supportive evidence to suggest that one agent should be adopted in favour of the other [197,248,249]. A randomized trial found that oral fingolimod (a disease-modifying drug used in multiple sclerosis) was not better than the placebo for the treatment of CIDP [250]. PE has been found to provide only short-term benefits and deterioration thereafter [244]. Recommendations by the European Academy of Neurology/Peripheral Nerve Society includes the use of IVIG or corticosteroids as the initial treatment while plasma exchange is recommended if IVIG or corticosteroids are ineffective. The guidelines also suggest using IVIG as first-line treatment for motor CIDP, and considering the use of IVIG, subcutaneous Ig or corticosteroids for maintenance therapy [210]. Treatment with IVIG is less frequently discontinued by patients than treatment with corticosteroid methylprednisolone during the initial 6 months of treatment [251]. A phase 3 randomized-controlled trial showed that subcutaneous Ig therapy was efficacious and well tolerated in CIDP patients as maintenance therapy [252]. A further follow-up study of these patients provided further evidence for its safety, tolerability and efficacy [253].

In 1982, Lewis et al. reported five patients with a sensorimotor mononeuropathy multiplex of subacute onset [254]. None of the patients had an underlying systemic disorder, and some patients responded to corticosteroid treatment. The involvement of the limbs was symmetrical, with sensorimotor features and generalized areflexia. The electrophysiology showed widespread conduction abnormalities as well as sensory abnormalities. The authors argued that the condition was immune-mediated, and was termed multifocal acquired demyelinating sensory and motor neuropathy (MADSAM) or Lewis–Sumner syndrome. Sural nerve biopsies revealed more abnormalities involving inflammatory infiltrates than those seen in MMN [255]. CSF protein levels tend to be more elevated than that observed in patients with MMN, and anti-GM1 antibodies are generally not detected. Patients respond to corticosteroids, unlike patients with MMN, and may also respond to intravenous immunoglobulin [255,256,257].

## 4. Neuropathies with Conduction Blocks

MMN with conduction blocks is a slow progressive, immune-mediated motor disorder. The early reviews [258,259] described patients presenting with a multifocal lower motor neuron disorder in the absence of objective sensory findings (Table 1). Patients present with a progressive asymmetric lower motor neuron syndrome that generally affects the upper extremities first. The reflexes tend to be preserved, although some focal loss of reflexes and global areflexia may occur when profound weakness develops. The diagnosis of MMN and differentiating it from other types of neuronal diseases such as amyotrophic lateral sclerosis (ALS) is vital, as MMN has a better prognosis with treatment [260,261]. Clinically, the progression of MMN is slower as compared to ALS. Furthermore, muscle weakness is typically in the distribution of motor nerve as compared to ALS where the entire myotome is affected, and nerve conduction study results are normal. In addition, upper motor neuron signs are absent in MMN, along with the sparing of cranial and respiratory muscles [262]. Nerve ultrasound and MRI are also useful in differentiating between MMN and ALS, where focal nerve enlargement can be detected in MMN [263,264]. Nerve hypertrophy may be observed radiologically [265] (Muley et al., 2012). The electrophysiological criteria for diagnosing MMN include findings of segmental demyelination and conduction blocks [266,267] and are confined to motor nerve fibres. CSF protein tends to be normal and titres of IgM antibody to GM1 ganglioside are elevated [268,269]. It is thought that anti-GM1 antibody is primarily responsible for causing conduction blocks. Other proposed mechanisms include the blockade of sodium channels, or hyper-polarization of nerve fibre membranes mediated by potassium channels, thus increasing a nodal threshold to the extent that this results in conduction blocks [270]. In vitro studies have shown that anti-GM1 antibodies also activate the complement system [132]. The analysis of serum for IgM antibody to GM1 ganglioside titres have been found to be efficient for diagnosis [271]. The activation of the classical pathway of complement is observed in the serum of patients with MMN, with significant association between muscle weakness and the loss of axons observed with higher complement-activating capacity of anti-GM1 IgM antibodies [272]. An in vitro study in a pluripotent stem cell model of MMN showed a strong expression of CD59 (a complement regulator), and this study also showed that a novel monoclonal antibody, which targets C2, was found to inhibit complement activation downstream of C4, being induced by patient-derived anti-GM1 antibodies that bind to motor neurons [273].

The disorder is slowly progressive and responds to treatment with IVIG [274,275]. The recommended dosage of IVIG over 2–5 days is a total of 2 g/kg [276]. The mechanism of action could be due to its effect on the reduced classical pathway of complement activation [277] or the inhibition of deposition of complement proteins at the node of Ranvier [278]. The inhibition of complement activity with eculizumab has also been found to be slightly effective [279]. Harbo et al. [280] found that subcutaneous immunoglobulin along with IVIG may also be beneficial.

## 5. Paraneoplastic Neuropathies

Neuropathies in cancer are often due to a combination of factors including the direct compression or infiltration by tumour, damage secondary to metabolic dysfunction, nutritional deficiencies, chemotherapy-associated neurotoxicity, or secondary to a paraneoplastic disorder. Table 2 summarizes the salient features regarding the latter. Paraneoplastic neuropathies are autoimmune, a result of tumour cells expressing onconeural antigens that mimic molecules expressed on the surface of neurons and mediated by an anti-tumour immune response, which attacks neurons co-expressing the related antigen. The immune response often keeps the tumour biologically benign (in size and non-metastatic) and therefore difficult to diagnose. The clinical presentations of neuropathies may vary depending on specific antibodies induced by specific tumours [281]. Patients commonly present with a subacute sensory neuronopathy, secondary to damage to the dorsal root and Gasserian ganglia and dropout of primary sensory neurons [282,283]. The infiltration consists of T and B cells, plasma cells and macrophages, usually in a perivascular distribution [284]. There is profound loss of myelinated fibres in peripheral sensory nerves, nerve roots and dorsal columns, probably secondary to the loss of dorsal root ganglia neurons. Neurophysiology often demonstrates a mixed sensorimotor neuropathy with the motor nerve conduction studies showing less significant abnormalities than the sensory ones. A large proportion of these patients are positive for anti-Hu IgG antibodies (ANNA-1). Most of these patients will harbour a small cell lung carcinoma [285,286]. Alternative antibodies, which are yet unidentified or not fully recognised, might be associated. These may include serum factors that inhibit Trk-neurotrophin receptor activity and that inhibit neurite outgrowth [287].

A smaller number of patients with a subacute sensory neuronopathy may have antibodies to other neuronal antigens, for example, anti-CV2 (CRMP-5), directed towards proteins expressed by neurons and oligodendrocytes [92], anti-amphiphysin antibodies [288], anti-Ma 1 antibodies [289], or ANNA-3 antibodies [290]. In some patients, gangliosides may represent the onconeural antigen and their expression on neoplastic tissue may elicit an autoimmune response, also targeting neural structures [291]. Patients, who are diagnosed early and treated for their malignancy, understandably show a better prognosis, but in most, the condition tends to stabilise probably because neuronal damage and loss are already established. Patients often benefit further if they are treated early and simultaneously with prednisolone or IVIG [292,293]. Patients with anti-Hu antibodies may also present with paraneoplastic autonomic insufficiency [294], highlighting the pan-neuronal activity of anti-Hu antibody activity, as demonstrated by the staining of sympathetic ganglia and myenteric plexus. More acute presentations, some resembling a GBS-like picture, generally occur in the context of lymphomatous disease [295]. Anti-disialosyl IgM occurring in the context of malignant lymphoma, often results in a sensory-ataxic neuropathy, with associated degeneration of the dorsal spinal columns, but motor nerve involvement also does occur [296].

Monoclonal gammopathies are a spectrum of disorder of clonal plasma cells, which includes monoclonal gammopathy of undetermined significance (MGUS) and Waldenstrom Macroglobulinemia (WM). The characteristic feature of monoclonal gammopathies is the secretion of monoclonal Ig known as monoclonal or M protein. Patients with MGUS are not only at a higher risk of developing peripheral neuropathy, but peripheral neuropathy is also under-recognised in MGUS patients [297]. IgM M proteins are associated more commonly with peripheral neuropathy as compared to IgG or IgA M proteins, and clinical presentation consists of distal, acquired, demyelinating, symmetric neuropathy with M protein [298]. Anti-myelin-associated glycoprotein (MAG) antibody is found in nearly half of MGUS patients with neuropathy [299].

Polyneuropathy, organomegaly, endocrinopathy, M protein, and skin changes (POEMS) syndrome is also a plasma cell disorder, in which neuropathy is characterised by progressive and often painful, chronic demyelinating sensorimotor polyneuropathy, which may affect lower limbs earlier and more severely than upper limbs [300]. IgA or IgG M proteins are more common in POEMS [296]. The cause for neuropathy is largely less well understood, and treatment with radiotherapy or chemotherapy usually improves neuropathy [296]. Peripheral neuropathy is also common in WM patients, with nearly one in four patients developing neuropathies [301]. Patients usually develop distal, sensory neuropathy similar to that in POEMS syndrome [302]. Treatment involves chemotherapy with rituximab and supportive therapy [300].

Anti-MAG neuropathy occurs when antibodies attack MAG proteins, and this type of neuropathy is observed in about 5% of patients with CIDP-like features [303]. Patients usually present with chronic, slowly progressing neuropathy that affects lower limbs initially, and is often associated with gait imbalance, sensory ataxia and tremors [300]. Rituximab has been shown to be the most effective treatment, with about one-third of patients responding well to rituximab, one-third showing some benefits while one-third not responding at all [251,304].

## 6. Conclusions and Perspectives

Dysimmune neuropathies are a heterogenous group of diseases with varying clinical presentation and pathophysiology. In some cases, these can be classified according to the presence of the different anti-glycoconjugate autoantibodies, which can be detected using ELISA or immunoblotting assays. These techniques, however, have existing limitations, which are mainly secondary to cross-reaction of the common glycosylate epitope. Further sub-classification into clinically distinct entities helps recognize those individual patient groups, who are more likely to respond to treatment modalities.

There is sufficient clinicopathological evidence to support an inflammatory immune-mediated aetiology even in metabolic conditions such as diabetes. However, the clinical response to immunomodulatory treatments, for instance in diabetic amyotrophy, is somewhat limited. Autoantibodies associated with connective tissue disorders or with cancer are generally indicators of the aetiology of the neuropathy. Early treatment of the primary condition may determine reversibility in some cases and improve the prognosis of the neuropathy.

There is a continuous attempt to classify neuropathies, based on our understanding of the pathophysiological mechanisms, and which is based on electrophysiological, immunological and pathological studies quoted in the literature. The electrophysiological findings are often paramount in the diagnosis of treatable, sometimes reversible conditions and in the classification of the neuropathy. In clinical practice, however, this is sometimes performed with difficulty since these conditions often exist as a wide-spanning spectrum with ill-defined boundaries. This often determines what clinicians communicate to their patients about their condition, and the clinician’s choice of treatment both in the acute phase and as maintenance.

## Figures and Tables

**Figure 1 ijms-24-07288-f001:**
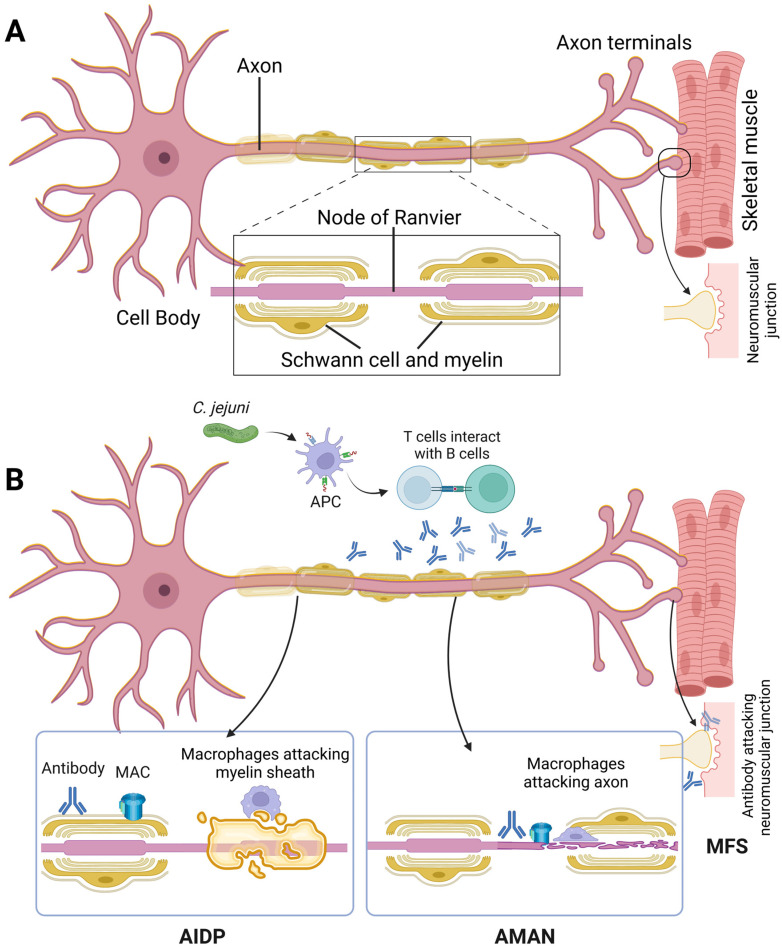
(**A**) A typical motor nerve is shown here. The nerve axon is covered by Schwann cells, which form the myelin sheath. This myelin sheath is discontinuous and contains gaps called as the node of Ranvier. The action potential travels from node to node. (**B**) In AIDP, the myelin sheath is affected; in AMAN and AMSAN, the axon is affected, while in MFS, the neuromuscular junction area is affected. Abbreviations: APC: antigen-presenting cell; MFS: Miller–Fisher syndrome.

**Figure 2 ijms-24-07288-f002:**
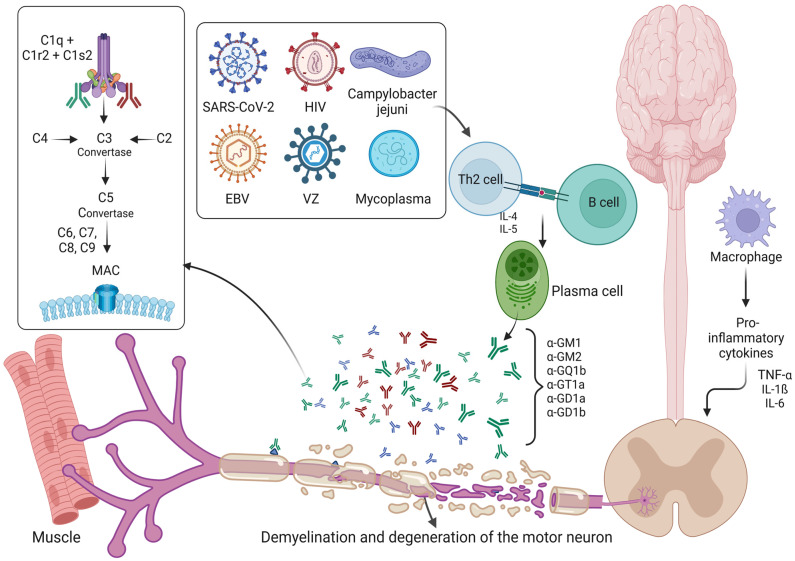
Immunopathological mechanisms involved in Guillain–Barre syndrome. A schematic diagram shows the various immunopathological mechanisms observed in Guillain–Barre syndrome. Preceding infection is usually observed, and this can involve various bacterial, viral and fungal pathogens. The subsequent activation of plasma cells and the molecular mimicry involving gangliosides, leading to destruction of myelin and axons, are shown. Proinflammatory cytokines too play a role in disease pathogenesis. The complement system is activated and is significantly involved in disease pathogenesis. Abbreviations: HIV: Human immunodeficiency virus; VZ: Varicella Zoster virus; MAC: Membrane attack complex.

**Table 1 ijms-24-07288-t001:** Summary of the types of immune-mediated neuropathies and their key features.

Neuropathy	Variants	Ab or Immune-Complex Targets	Antibodies	Preceding Infections/Associations	Treatment Options	Diagnostic Criteria
Guillain–Barre syndrome	(1) Acute inflammatory demyelinating polyradiculoneuropathy (AIDP)	Schwann cell plasmalemma	none	*Campylobacter jejuni*, Mycoplasma, EBV, CMV, Varicella zoster, HIV, and Hodgkin’s lymphoma	Intravenous immunoglobulin Plasma exchange	(1) Clinical presentation plateaus at 4 weeks (2) Increased CSF protein (3) Acellular CSF (4) Neurophysiological findings: (i) demyelination ± axonal loss (ii) prolonged distal motor latency (iii) prolonged F wave latency (iv) Temporal dispersion of compound muscle action potentials (v) Conduction block
	(2) Acute motor axonal neuropathy (AMAN)	Axolemma	Anti-GM1, Anti-GM2, Anti-GQ1b, Anti-GT1a, Anti-GD1a, Anti-GD1b			
	(3) Acute motor and sensory axonal neuropathy (AMSAN)	Axolemma	As in (2)			
	(4) Sensory variant	Axolemma	As in (2)			
	(5) Miller–Fisher	Cranial motor nerves	Anti-GQ1b			
CIDP	Relapsing	As for GBS	Anti-gangliosides as for GBS (presumed to be involved in CIDP as well, but not yet documented). In a small proportion of patients, antibodies against proteins located at the node of Ranvier are detected (antibodies target neurofascin, contactin and contactin-associated protein 1)	Diabetes mellitus; Connective tissue diseases; infections; dysproteinaemias	Intravenous immunoglobulin; Plasma exchange; corticosteroids; immunosuppression	Same as in GBS but clinical presentation is more progressive and prolonged
	Chronic monophasic					
	Slow progressive					
Neuropathy with conduction block	MMN	Probably the nodes of Ranvier are a major target but also the axolemma and Schwann cell plasmalemma	Anti-GM1	None	Intravenous immunoglobulin	Multifocal conduction abnormalities with conduction blocks and no sensory features electrophysiologically
	MADSAM		Negative for anti-GM1	None	Intravenous immunoglobulin and corticosteroids	Multifocal conduction abnormalities with conduction blocks and sensory abnormalities

**Table 2 ijms-24-07288-t002:** Paraneoplastic neuropathies and their key features.

Paraneoplastic Neuropathy	
Variants	○ Subacute sensory neuronopathy ○ Motor neuropathy ○ GBS-like neuropathy sometimes ○ Autonomic neuropathy
Antibody or immune-complex targets	○ Dorsal root ganglia and sympathetic ganglia ○ Axolemma mainly ○ Myenteric plexi
Antibodies	○ Anti-Hu IgG antibodies (ANNA-1 and ANNA-3) ○ Anti-CV2 (CRMP-5) antibody ○ Anti-amphiphysin antibodies ○ Anti-Ma 1 antibodies
Preceding infections/associations	○ Small cell or non-small cell lung carcinoma ○ Breast or ovarian cancer ○ Prostate cancer ○ Neuroblastoma ○ Hodgkin’s lymphoma ○ Thymoma
Treatments	○ Of the malignancy itself ○ Corticosteroids ○ Plasma exchange ○ Immunoglobulin ○ Other immunosuppression with caution
Diagnostic criteria	○ Often axonal damage electrophysiologically ○ Indolent course ○ Presence of antibody even in the initial absence of tumours

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
