# Peer review of "Immune-Mediated Neuropathies: Pathophysiology and Management"

_ijms, 2023, doi:10.3390/ijms24087288_

Round 1

Reviewer 1 Report

The introduction is written confusingly and does not focus well on the purpose of the manuscript, ie immune-mediated neuropathies and the importance of knowing the pathogenesis and early diagnosis because they are potentially treatable. it should be rewritten.

For example: "This review...aims...in some of these conditions". This is wrong. This paper does not speak of metabolic neuropathies but of immune-mediated forms.

An introductory paragraph is missing that explains to the non-expert reader the classification of clinical and neurophysiological immune-mediated neuropathies.

There is confusion in the GBS paragraph. Sections 2.1 and 2.2 refer to the electrophysiological classification of GBS. Then section 2.3 refers to a clinical variant of GBS. And why are the other clinical variants not described?  Authors need to be clear about clinical classification with its various variants and electrophysiological definitions.

Paragraphs 2.4 and 2.5 appear to be detached from the remaining paragraphs and should be inserted in another way.

A descripion and a comment on the recent association between GBS and COVID-19 and its pathogenesis could be indicated in this paragraph.

The concept of node-paranodopathies is limited to a few lines in the paragraph of the CIDP while it is instead a current topic and deserves a more complete and extensive space.

Both in the GBS and in the CIDP paragraph, the part dedicated to the treatment is very short. However, the review is entitled "pathophysiology and management" therefore the part dedicated to management cannot be limited to a few lines on current therapy (which is already well known) but must consider all aspects of taking charge of the patient.

I do not agree that MADSAM is discussed in the paragraph of the MMN. It should be discussed with the CIDP.

The chapter on paraneoplastic neuropathies is also too small.                  Some other immune-mediated neuropathies could be discussed here. Otherwise, In which chapter would the authors include other immune-mediated neuropathies that they did not discuss such as neuropathies that occur with monoclonal gammopathy, polyneuropathy-organomegaly-endocrinopathy-M-component-skin changes syndrome (POEMS), cryoglobulinemia type I and glycoprotein antimyelin-associated (MAG) neuropathies and Waldenström disease?

Author Response

Thank you for the helpful feedback and comments to improve the article. Extensive changes and additions have been made, as per below. We have also included two new figures to the manuscript.

  • We have now changed and simplified the Introduction section.
  • A section on clinical and neurophysiological classification has also been added. 
  • Paragraphs 2.4 and 2.5 have now been amalgamated as one. A description on association of GBS and COVID has been included under this section as well. 
  • Nodo-paranodopathies section has been expanded. 
  • Management section too has been bolstered as suggested. 
  • MADSAM has now been moved under CIDP section. 
  • And as suggested, other immune-mediated neuropathies have now also been added. 

Thank you again.

Yours sincerely,

Authors.

Author Response

Thank you for the helpful feedback and comments, to improve the article.  We have incorporated suggestions as per below. We have also included two new figures to the manuscript.

  1. As suggested, information about nerve conduction studies, have now been added. Further details about macrophages and cytokines too have been added.  
  2. Details about clinical course of GBS are now addressed, and perspective on similarities in plasma exchange and IVIG has also been added. 
  3. Information about TNFα antagonists and their trials have been added. 
  4. Gangliosides are now discussed alongside their association with clinical presentation
  5. A section on nodo-paranodopathies has now been added. 
  6. Difficulties in electrophysiology diagnosis in CIDP has now been addressed, and a note on utility of ultrasound and MRI has been added as well. 
  7. Done 
  8. Discussion on distinguishing between MMN and ALS added. 
  9. Now addressed as well. 

Thank you again.

Yours sincerely

Authors.

Reviewer 3 Report

The manuscript here summarized the current research status and potential immunological therapies for various peripheral neuropathies. This review did a great job at discussing variants of the peripheral neuropathy disorder. The manuscript is well-organized, with a clear structure and flow. The information is presented in a logical and easy-to-follow manner, making it a pleasure to read.  I have a minor comment that could be easily dealt with.

1. The tables 1 and 2 very nicely summarized the different neuropathy disorders. However, these tables were not frequently referenced in the text. The authors could refer to the tables more often to give more context to the readers when going through the manuscript.

Author Response

Thank you very much for the feedback and comments. We have now cited the Tables more often in the text. We have also included two new figures to the manuscript.

Thank you again.

Yours sincerely

Authors.

Round 2

Reviewer 2 Report

The revised manuscript is better balanced and much improved. The addition of the uncertainty in electro- diagnosis of CIDP and the comments on the role of MRI and ultrasound are a welcome improvement. The section on "Neuropathies with conduction block", and the "Conclusions and perspectives" are very good. However: a few relatively small points should be addresed :

1. The text in 474-476 and the part of Table 1 dealing with targets of antibodies in CIDP are overstated. Antibodies have been documented and their targets identified in a very small proportion of CIDP cases. I suggest to add the word "presumed" or "likely" in line 474-5 to indicate that antibodies against myelin sheaths are "presumed" or "likely" to be involved. Similarly, in the Table, it should be clear that antibodies against gangliosides are presumed to be involved but not yet documented. Line 547, indicting that "only a small proportion of CIDP patients" have antibodies against glycolipids. Similarly in lines 552-553, it should be noted that antibodies against nodal and paranodal proteins have been identified in only a small proportion of CIDP patients. These antibodies should also be noted in Table 1, again with the caveat that they have only been identified in a small proportion of patients. 

2. The sentence in lines 646 to 649 describing the findings of Budding et al in MMN appears to be missing a phrase between the words "C2" and "showed strong expression". Expression of the complement regulators is not likely caused by the experimental antibody against C2. 

3. The section on "Neuropathies in cancer" is good, and is completely appropriate for this special issue. However, the meaning of the sentence on lines 668-669: "the presence of antibody is merely a marker of a specific cancer" leads to the inference that the antibody does not have a pathogenic role in the neuropathy. This should be clarified, perhaps by suggesting that the presentations of specific neuropathies varies with the specific antibodies induced by specific tumors. 

Author Response

Thank you very much for the feedback and helpful comments. 

1. “presumed to be” has now been added. In Table 1 too, addition of “presumed to be involved but not yet documented” added. 

Further changes made to add that only a small proportion of patients have antibodies against nodal and paranodal proteins in CIDP. Similar addition and changes made in Table 1 as well. 

2. This has now been corrected. Thank you. 

3. Addition of the phrase suggested has now been added. 

Thank you again. 

Yours sincerely, 

Authors